# Nd: YAG Pulsed Laser Dissimilar Welding of UNS S32750 Duplex with 316L Austenitic Stainless Steel

**DOI:** 10.3390/ma12182906

**Published:** 2019-09-09

**Authors:** Carla Gabriela Silva Leite, Eli Jorge da Cruz Junior, Mattia Lago, Andrea Zambon, Irene Calliari, Vicente Afonso Ventrella

**Affiliations:** 1Department of Mechanical Engineering, Sao Paulo State University, Av Brazil Centro 56, 15385-000 Ilha Solteira SP, Brazil; 2Instituto Federal de Educação, Ciência e Tecnologia de São Paulo, Av. Jerônimo Figueira da Costa 3014, 15503-110 Votuporanga SP, Brazil; 3Department of Industrial Engineering, University of Padua, Via Marzolo 9, 35131 Padova PD, Italy (M.L.) (A.Z.) (I.C.)

**Keywords:** Nd: YAG, laser welding, dissimilar material, duplex steel, austenitic steel, microstructure, microhardness

## Abstract

Duplex stainless steels (DSSs), a particular category of stainless steels, are employed in all kinds of industrial applications where excellent corrosion resistance and high strength are necessary. These good properties are provided by their biphasic microstructure, consisting of ferrite and austenite in almost equal volume fractions of phases. In the present work, Nd: YAG pulsed laser dissimilar welding of UNS S32750 super duplex stainless steel (SDSS) with 316L austenitic stainless steel (ASS), with different heat inputs, was investigated. The results showed that the fusion zone microstructure observed consisted of a ferrite matrix with grain boundary austenite (GBA), Widmanstätten austenite (WA) and intragranular austenite (IA), with the same proportion of ferrite and austenite phases. Changes in the heat input (between 45, 90 and 120 J/mm) did not significantly affect the ferrite/austenite phase balance and the microhardness in the fusion zone.

## 1. Introduction

Recently, industries have been incorporating many kinds of materials in their products to reduce costs and improve performance. Thus, the use of new technologies for welding dissimilar materials in industrial production is increasing rapidly. Laser welding has many advantages and limitations when compared with other welding processes. The keyhole mode of welding uses reduced energy transfer to the material, resulting in a very narrow heat affected zone (HAZ) with small distortions and low residual stress. In laser welding with the keyhole mode, the beam is focused on the surface of the material, with power density high enough to start vaporization. A deep and narrow vapor cavity, the keyhole, is then formed, which allows multiple internal reflections of the beam and, consequently, very efficient power deposition. The keyhole is surrounded by molten material, forming a fusion zone, and by an adjacent solid distinct region, which is the HAZ [1,2,3,4,5].

Duplex stainless steels (DSSs), such as Alloy 2507 (UNS S32750), are frequently used in aggressive corrosion environments, provided that service temperatures are below 350 °C. The chemical composition of duplex steels contains 24%–26% Cr, 5%–8% Ni, 3%–5% Mo and 0.2%–0.3% N. Duplex stainless steels have excellent resistance to pitting and stress corrosion cracking. Extensive control of the austenite and ferrite phase balance (50%–50%) in the weld metal is very important for good corrosion resistance. Due to the thermal cycle experienced during welding with high power density sources, the optimal ferrite/austenite ratio is impaired, and the phase balance can be restored by a post weld heat treatment (PWHT). For field welding, as in the oil and gas industry, PWHT is not always possible due to the large dimensions of components, which may compromise the corrosion resistance of the welded structure. Thus, to improve the corrosion resistance of the weld deposit, one of the solutions would be the addition in the weld bead of elements promoting the gamma phase, such as Ni [6,7,8,9].

Due to the high nitrogen and low carbon contents, DSSs exhibit good laser weldability. In a general way, the fusion zone microstructure presents more ferrite content than that predicted by the Schaeffler diagram. Normally, the fusion zone microstructure contains between 15% and 25% austenite. The high cooling rate and rapid solidification in the laser welding process does not affect ferrite formation, but it suppresses the solid state ferrite to austenite phase transformation. The microstructural balance of austenite and ferrite with approximately 50%–50% is reached with post weld heat treatment in the range of 1050–1100 °C, and good stress corrosion properties are obtained. Annealing and quenching encourages austenite formation in autogenous welds. The addition of alloys with high Ni content may also be used to increase the austenite volumetric fraction. After the welding, the fusion zone hardness typically increases up to 350 HV [10,11,12,13,14].

Steels of the austenitic group exhibit good resistance to corrosion and oxidation at temperatures up to 650 °C or higher. These steels also exhibit excellent ductility and toughness in this temperature range. Corrosion and oxidation resistance is imparted primarily by a high chromium content, generally higher than 16 wt.%. The addition of austenite-stabilizing elements, primarily carbon and nickel and sometimes nitrogen, promotes an austenitic structure over a wide range of temperatures. In some alloys, austenite is stable from room temperature to the melting temperature range. The predominance of an austenitic structure in these steels gives rise to their excellent ductility and toughness [15,16,17,18,19].

The wide utilization of plain austenitic stainless steels, due to excellent corrosion resistance and good cost-effectiveness, has stimulated research about their applications in many situations and environments, so that much information is available concerning their shaping, welding (even with high power density facilities) and, eventually, post weld heat treatments. Currently, superaustenitic stainless steels are being used because they can bridge between relatively cheap austenitic stainless steels (ASSs) and expensive Ni base super alloys, when high corrosion properties are required at moderately high temperatures [20,21,22,23].

The fusion zone chemical composition of welds of dissimilar steels has an intermediate composition between the different base metals. This has a significant influence on the weld metal microstructure, its properties and its weldability. It is important to know how variations in processing parameters affect the fusion zone composition for dissimilar steels welding applications [24,25,26].

In this research, both the super duplex stainless steel UNS S32750 and the austenitic stainless steel 316L, Nd: YAG pulsed laser welded under different heat inputs of 45, 90 and 120 J/mm, were studied, and the microstructure/mechanical properties relationship was analyzed.

## 2. Materials and Methods

AISI 316L austenitic stainless steel (ASS) with UNS S32750 super duplex stainless steel (SDSS) were the base metals used in this research, with their composition (wt.%) shown in Table 1. The base metals were shaped to sizes of 1.5 mm × 50 mm × 120 mm. The laser welding studies were conducted analyzing the relationships between pulsed laser welding parameters, heat input, chemical composition, mechanical and microstructural properties. Before welding, the base metals were cut using the electro-discharge machining (EDM) process to ensure no root opening in the region of the butt joint faces. Samples were welded with single pass-double size with a pulsed Nd: YAG laser machine, model UW-150A, with a 150 W maximum mean power. The laser beam with pulse energy set at 10 J was focused on the surface of the base metal (BM), and the welding speed was fixed at 1.0 mm/s. The frequencies of the pulses used were 4.5, 9.0 and 12.0 Hz. Thus, the controlled parameter in this process was the heat input. The joints were laser-welded in an argon atmosphere at 15 l/min. Table 2 shows the welding parameters that were used. Figure 1 shows a schematic of the dissimilar butt joint (Figure 1a) and an optical macrograph of the dissimilar steels pulsed laser welded joint (Figure 1b).

Beraha reagent (85 mL H_2_O–15 mL HCl–1 g K_2_S_2_O) was used for etching and revealing the macro and microstructures. The austenite/ferrite ratio in the fusion zone was calculated using SEM images (Carl Zeiss EVO LS15, Munich, Germany) and free image software (Image J, version 1.51k, National Institutes of Health, MD, USA).

Vickers microhardness tests were performed at 0.5 mm from the top surface on the cross-section of the dissimilar steels welded joint. Standard 100 gf was used for a residing time of 10 s and intervals of 0.1 mm. The tensile tests were performed following ASTM E8M-04 Standard. Due to the small plate thickness, sub-sized specimens were used.

## 3. Results

### 3.1. Macro and Microstructures of the Welded Joint

Figure 2 shows the microstructures of the base materials with 1.5 mm thickness. The microstructures show the presence of balanced austenite and ferrite phases on 32750 duplex steel and an austenitic microstructure on 316L austenitic stainless steel. The weld beads showed the characteristic morphology of pulsed laser welding, which is a sequence of ripples due to the partially superimposed laser beam shots (see Figure 3). The effect of dissimilar base materials with different compositions on the microstructure of joints welded by the Nd: YAG pulsed laser was investigated.

The microstructures of the transition regions between the base metals, UNS S32750 and AISI 316L, and the fusion zone are shown in Figure 4. In the fusion zone, austenite was formed following three modes: allotriomorphs at the prior-ferrite grain boundaries, Widmanstätten side-plates growing into the grain from the allotriomorphs’ grain boundaries, and intragranular austenite.

The quantitative effect of the high nickel content from the AISI 316L base metal in the fusion zone of the dissimilar steels welding process can be observed in Table 3, which shows the proportions of austenite and ferrite for all welding conditions and for the base metal (BM). The results were obtained by image analysis in several regions of the fusion zone (FZ) and base metal (BM).

The fusion zone micrographs of the weld beads for different heat input conditions (45, 90 and 120 J/mm, respectively) are shown in Figure 5. As already shown, it is possible to verify that different heat inputs did not significantly affect the amount and morphology of austenite.

### 3.2. Microhardness

The Vickers microhardness profile for dissimilar steels welded with 45, 90 and 120 J/mm is shown in Figure 6.

The mean microhardness profiles of the base metals, UNS S32750 and AISI 316L, were 292 ± 5.8 HV and 238 ± 3.2 HV, respectively. The fusion zones presented a predominantly duplex microstructure, and the microhardness in the fusion zones was higher than in the base metals, 329 ± 6.3 HV.

### 3.3. Tensile Strength

In the tensile tests of all samples, fracture occurred outside the fusion zone, always in the 316L austenitic stainless steel, as shown in Figure 7, with tensile stress of roughly 540 MPa.

## 4. Discussion

As reported by Mirakhorli et al. [9], the welding of duplex steel using a laser process produces a microstructure in the fusion zone with a low volumetric fraction of austenite in the form of allotriomorphs at ferrite grain boundaries and acicular intragranular austenite. The resulting microstructure is unbalanced, with the ferrite predominating. In the present research, it is possible to see the presence of an austenitic phase in the three forms, and in a greater volumetric fraction, resulting in a visibly balanced microstructure. The high nickel content from the AISI 316L base metal affected the austenite both qualitatively and quantitatively, modifying the morphology and volumetric fraction. The morphology of the formed austenite will be discussed later.

Increasing the nickel content in the fusion zone, the austenite proportion increased to approximately 52%. This effect is a consequence of more austenite-promoting elements in the fusion zone. As already mentioned, phase balance in SDSS is very important for the material to maintain its mechanical resistance and corrosion properties [27,28,29,30].

Regarding the morphology of the austenitic phase, it is known that the grain boundary and Widmanstäten austenite form at high temperatures, while the intragranular austenite forms at lower temperatures, so high cooling rates favor the formation of the latter. In SDSS weldments, austenite is formed by solid-state transformation, which is strongly affected by the cooling rates and heat input. Under high cooling rates, there is no time for completing the ferrite-to-austenite formation up to the optimal phase ratio, resulting in a predominantly ferritic microstructure. The great amount of intragranular austenite in all conditions is the result of the thermal sequence (in particular, the high cooling rate) of the laser process associated with the austenite promotion effect of the high nickel content from the AISI 316L austenitic base metal.

Comparing the resulting microstructures for the different conditions, it can be said that the effect of the chemical composition of the AISI 316L base metal, which contains large amounts of austenite-stabilizing elements, overlapped the effects of the thermal cycle once the same phase balance was reached with different heat inputs.

The feasibility of welding by pulsed Nd: YAG laser between a duplex and an austenitic stainless steel containing 10 wt.% Ni as the austenite-promoting element has been demonstrated.

Moreover, it has been proved that with the pair of dissimilar steels tested and with the adopted set of welding parameters, a satisfactory phase balance, very close to that of the duplex base metal, can be achieved directly from the cooling of the bead, without the need for post weld heat treatments.

## 5. Conclusions

In dissimilar Nd: YAG pulsed laser welding between UNS S32750 duplex steel and AISI 316L austenitic stainless steel, volumetric fractions of austenite and ferrite of approximately 50%–50% were obtained in the fusion zone, resulting in a balanced microstructure.Duplex microstructures in the fusion zone occurred as a consequence of the higher Ni content (austenite former) and lower Cr content (ferrite former) from the 316L austenitic stainless steel, when compared to the chemical composition of duplex steel.The hardness of the fusion zone was higher than that of the base duplex steel, since the different morphology and arrangement of the austenite along the fusion zone affected the hardness of the region.In all tensile tests, fracture occurred outside the fusion zone, always in the AISI 316L austenitic stainless steel base metal, with an average tensile strength of 540 MPa.Changes in the heat input did not significantly affect the ferrite/austenite phase balance and the microhardness of the fusion zone.Based on the outcomes of the present work, the Nd: YAG pulsed laser welding of dissimilar materials, namely, UNS S32750 super duplex stainless steel (SDSS) with AISI 316L austenitic stainless steel (ASS), is recommended.

## Figures and Tables

**Figure 1 materials-12-02906-f001:**
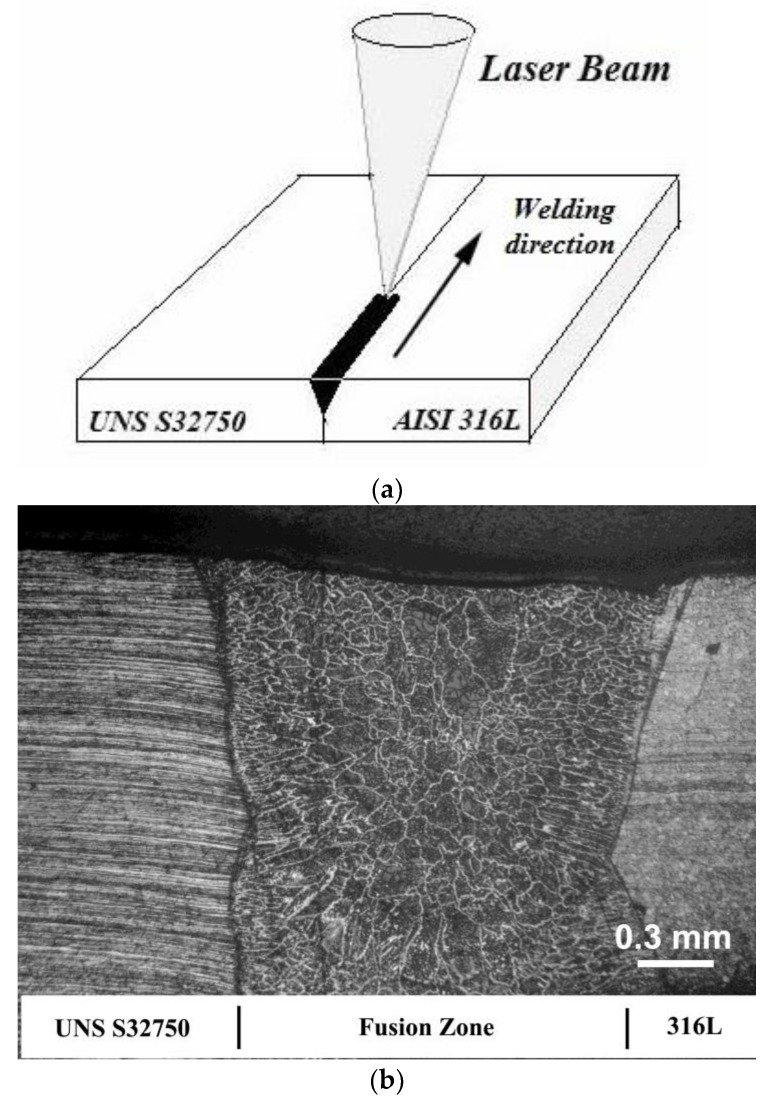
(**a**) Schematic of the dissimilar steels butt joint; (**b**) optical macrograph of the welded joint.

**Figure 2 materials-12-02906-f002:**
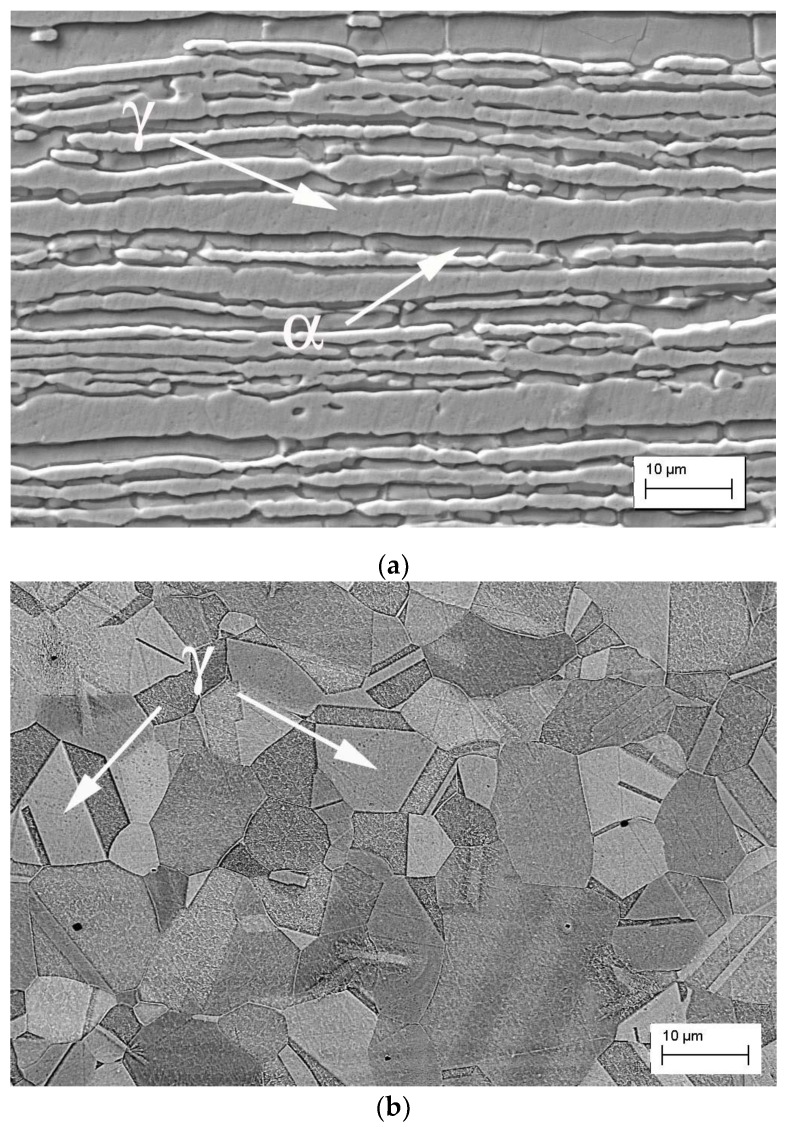
SEM micrographs of the base materials: (**a**) UNS S32750 duplex steel; (**b**) AISI 316L austenitic stainless steel. (γ-austenite, α-ferrite).

**Figure 3 materials-12-02906-f003:**
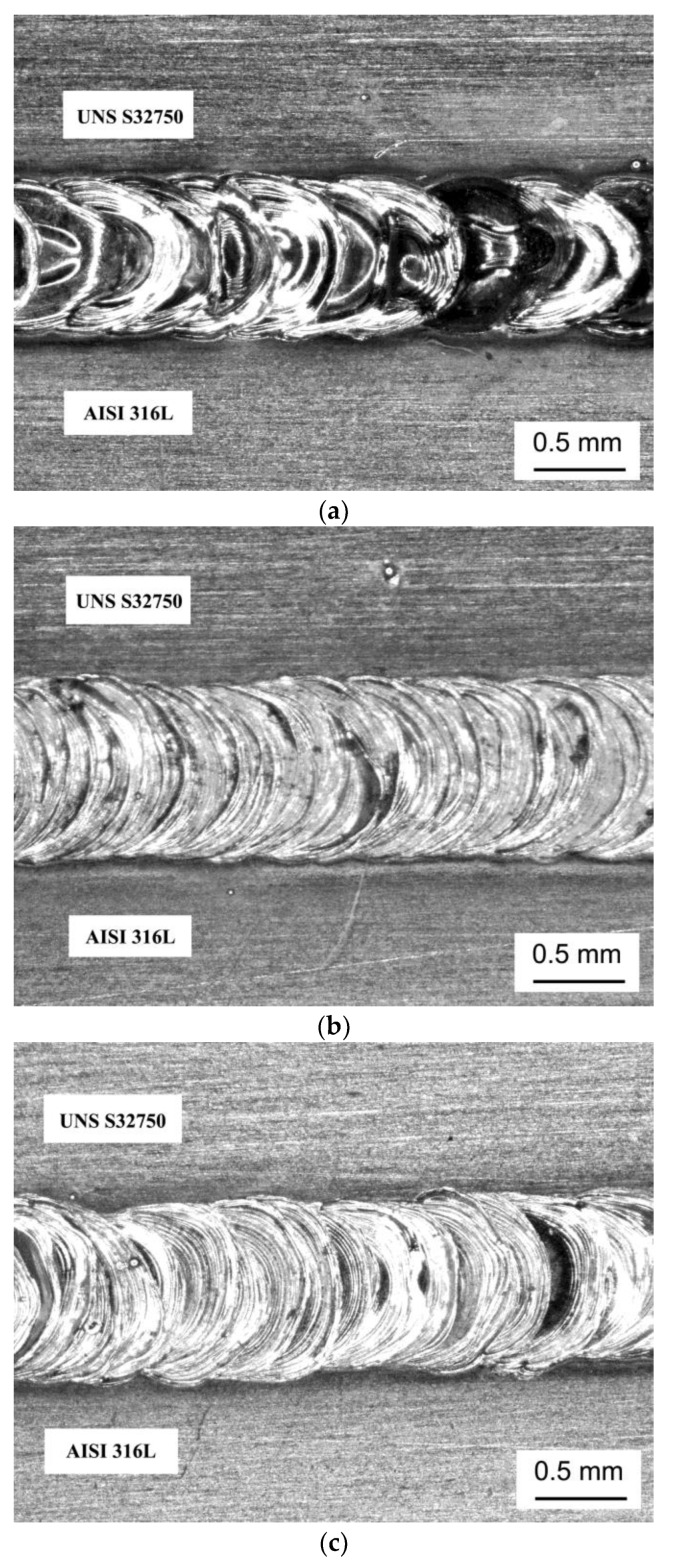
Top view of dissimilar welded specimens with: (**a**) 45 J/mm, (**b**) 90 J/mm and (**c**) 120 J/mm.

**Figure 4 materials-12-02906-f004:**
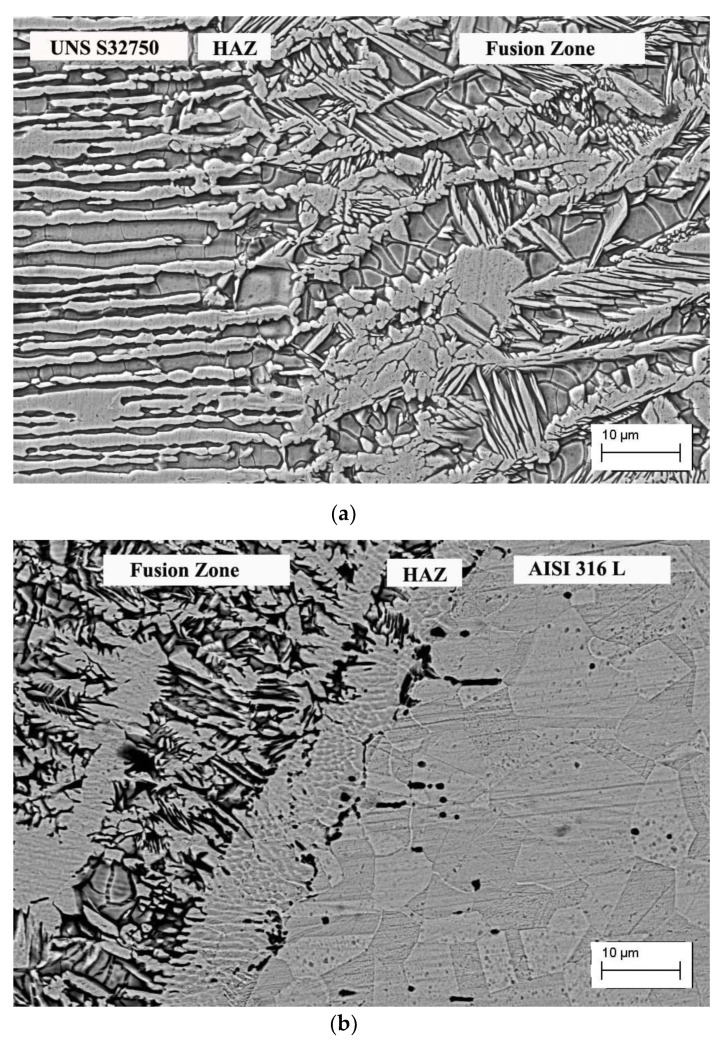
SEM micrographs of the transition region. (**a**) UNS S32750 base metal and fusion zone; (**b**) AISI 316L base metal and fusion zone.

**Figure 5 materials-12-02906-f005:**
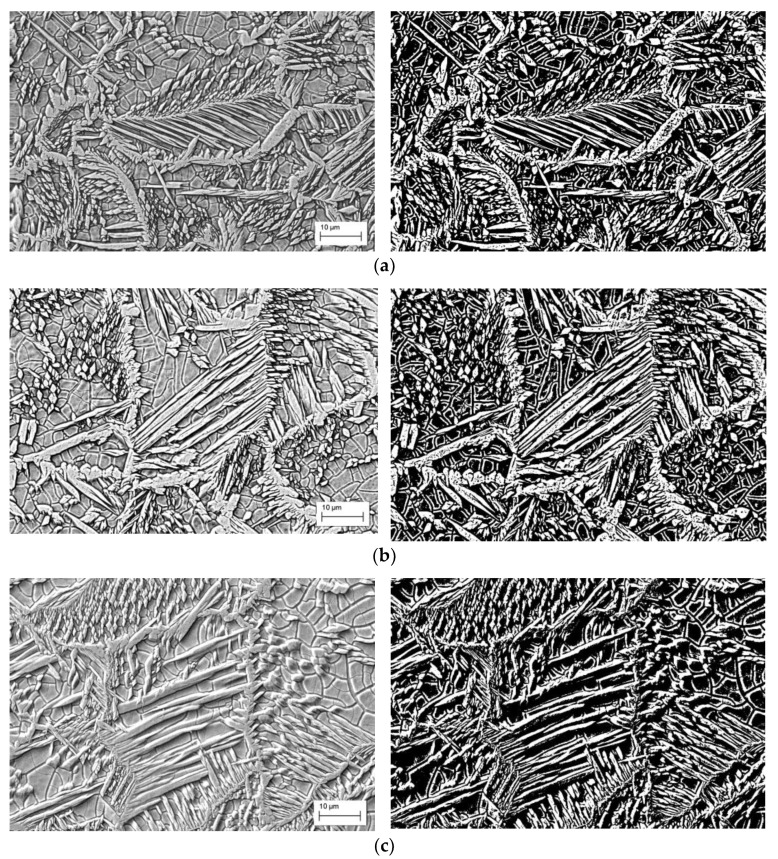
SEM micrographs and respective binary images of the fusion zone for: (**a**) 45 J/mm; (**b**) 90 J/mm; (**c**) 120 J/mm.

**Figure 6 materials-12-02906-f006:**
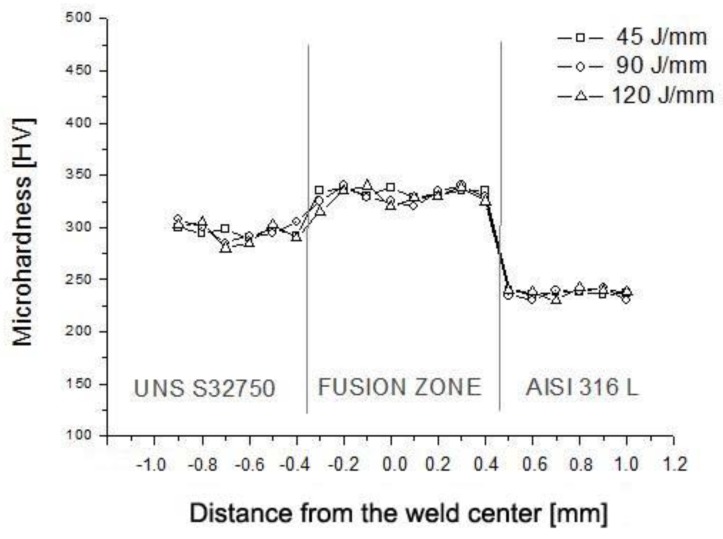
Microhardness profile along the dissimilar weld joints at different heat inputs.

**Figure 7 materials-12-02906-f007:**
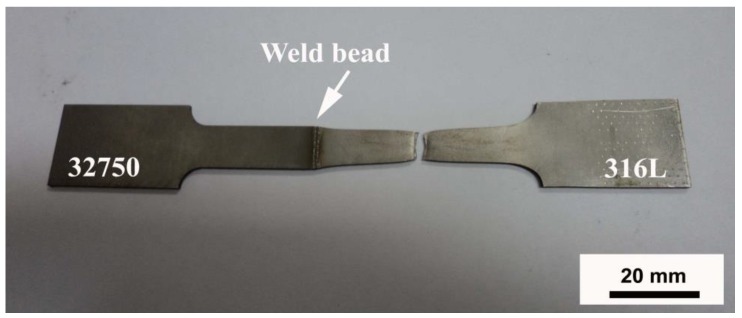
Tensile test specimen D3.

**Table 1 materials-12-02906-t001:** Chemical composition of the base metals (wt.%).

Base Metal	C	Si	Mn	Cr	Ni	Mo	S	P	Cu	N
32750	0.018	0.29	0.63	25.61	6.97	3.84	0.003	0.020	0.15	0.269
316L	0.020	0.35	1.5	16.25	10.11	2.04	0.011	0.030	0.19	-

**Table 2 materials-12-02906-t002:** Pulsed laser welding parameters.

Samples	Peak Power [kW]	Pulse Span [ms]	Pulse Energy [J]	Frequency [Hz]	Heat Input [J/mm]
D1	2	5	10	4.5	45
D2	2	5	10	9.0	90
D3	2	5	10	12.0	120

**Table 3 materials-12-02906-t003:** Phase proportions in the fusion zone.

Sample	Heat Input (J/mm)	Austenite %	Ferrite %
BM	-	49.8	50.2
D1	45	47.7	53.3
D2	90	48.2	51.8
D3	120	48.9	51.1

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
