# Peer review of "Nd: YAG Pulsed Laser Dissimilar Welding of UNS S32750 Duplex with 316L Austenitic Stainless Steel"

_materials, 2019, doi:10.3390/ma12182906_

Round 1

Reviewer 1 Report

The axis titles are not in the center of Fig. 6; The stress-strain curve of tensile test is missing; The font of words and the scale are too small for all figures; It would be better if the authors could provide some EBSD mappings; Please discuss the following paper in the introduction sector, which provided detailed discussion of friction stir processingZhu et al., Microstructures and mechanical properties of Al-Li 2198-T8 alloys processed by two different severe plastic deformation methods: A comparative study,2017

Author Response

English language was checked.

The axis-titles of Fig. 6 were put in the center of the graphic.

The authors didn’t put the stress-strain curve in the article because all the samples fractured in the austenitic stainless steel and no plastic deformation occurred in the duplex steel.

To increase the words and the scales size, without compromising the figure areas, the authors decided to increase the size of the figures.

About EBSD the authors are preparing these samples for other publication.

The authors decided not to compare, in the introduction section, a fusion process (laser welding) with a solid state welding (friction stir) understanding that they are different.

Reviewer 2 Report

Comments

Manuscript ID materials-576160

Title: Nd:YAG pulsed laser dissimilar welding of UNS S32750 duplex with 316L  austenitic stainless steel

Journal: Materials

The selected topic is current and  this article is     interesting.     The article contains many experiments,  but there is need  explain a What type of  the method on   determination  of the phase proportions in  the fusion zone   was used ( feritte and austenite- see tab 3).

Author Response

In the paragraph before tab.3 is specified “image analysis” as the method that was used in this article.